# The Dialogue study: Protocol for a randomized clinical trial evaluating the efficacy of virtual reality-based psychotherapy plus treatment as usual versus treatment as usual for eating disorders

Nina K. Hansen[1,2], Emma S. Ries[1], Thomas Ward[3,4], Valentina Cardi[5], Anne B. Christensen[2,6], Carsten Hjorthøj[7,8], Merete Nordentoft[8,9], Nadia Micali[6], Louise B. Glenthøj[1,2]*

1 VIRTU Research Group, Mental Health Center Copenhagen, Copenhagen University Hospital, Hellerup, Denmark, 2 Department of Psychology, University of Copenhagen, Copenhagen, Denmark, 3 Institute of Psychiatry, Psychology & Neuroscience, King's College London, London, United Kingdom, 4 South London & Maudsley NHS Foundation Trust, London, United Kingdom, 5 Department of General Psychology (IT), University of Padova, Padova, Italy 6 Center for Eating and Feeding Disorders Research, Mental Health Center Ballerup, Copenhagen University Hospital, Mental Health Services in the Capital Region, Ballerup¸ Denmark, 7 Department of Public Health, Section of Epidemiology (DK), University of Copenhagen, Copenhagen, Denmark, 8 Copenhagen Research Center for Mental Health – CORE, Mental Health Services in the Capital Region, Hellerup, Denmark, 9 Department of Clinical Medicine, University of Copenhagen, Copenhagen, Denmark,

* louise.birkedal.glenthoej@regionh.dk

## Abstract

### Background:

There is considerable interest in developing novel psychological interventions for eating disorders targeting characteristics potentially serving as maintaining factors in eating disorder pathology. An estimated 94% of patients with an eating disorder report a dominant internal voice commenting on weight and self-worth, often referred to as the 'eating disorder voice'. The experience of a more dominating 'eating disorder voice' has been linked to longer illness duration. Within psychotic disorders, an intervention termed AVATAR therapy, using computerized avatars, has proven effective in reducing the severity of the psychotic voice and the associated distress. Building on this evidence and the proof-of-concept for AVATAR therapy adapted to eating disorders, this study investigates an immersive version targeting eating disorder symptoms. In this adaptation participants engage with an avatar representing their inner eating disorder voice in virtual reality.

### Methods:

The study is designed as a randomized parallel-group superiority clinical trial. A total of 96 patients with an eating disorder will be allocated to either seven sessions of

**Data availability statement:** All data are in the manuscript and/or Supporting information files.

**Funding:** This project is supported by: - Independent Research Fund Denmark(reference-number: 2096-00078B, recepient LBG), Phone +45 35446800, dff@ufm.dk, https://dff.dk/en - Research Fund of the Mental Health Services – Capital Region of Denmark (recipient LBG), Phone +45 38665000, pure@regionh.dk, https://research.regionh.dk/en/organisations/region-hovedstadens-psykiatri .The funders had no role in the study design, data collection and analysis, decision to publish, or preparation of the manuscript.

**Competing interests:** The authors have declared that no competing interests exist.

**Abbreviations:** BAVQR, Revised Beliefs About Voices Questionnaire; BRIEF, Behavior Rating Inventory of Executive Functioning; BSQ-34, Body Shape Questionnaire; CIMT, Center for IT og Medicoteknologi; CSQ, Client Satisfaction Questionnaire; CTQ, Childhood Trauma Questionnaire; DAS-24, Dysfunctional Attitude Scale; Dflex, Detail and Flexibility Questionnaire; DMC, Data Monitoring Committee; DSM-IV, Diagnostic and Statistical Manual of Mental Disorders, Fifth Edition and International Statistical; EDQLS, Eating Disorder Quality of Life Scale; EAVE-Q, Experience of An Anorexic Voice Questionnaire; EDE-Q, Eating Disorder Examination Questionnaire; ERQ, Emotion Regulation Questionnaire; GSE, The General Self-Efficacy Scale; HADS, The Hospital Anxiety and Depression Scale; ICD-10, Classification of Diseases and Related Health Problems, 10th Revision; IDEA, Identity and Eating Disorder Questionnaire; MINI, Mini International Neuropsychiatric Interview; MPS, Multi Modal Presence Scale; NEQ, Negative Effects Questionnaire; P-CED, Pros and Cons of Eating Disorders Questionnaire; PI, Principal Investigator; PMG, Project Management Group; PSYRATS-AH, the Psychotic Symptoms Rating Scale, Auditory Hallucinations; QPR-15, Questionnaire about Process of Recovery; RCT, Randomized Clinical Trial; SAP, statistical analysis plan; SCS, The Self-Compassionate Scale; SD, Standard Deviation; SOCQ-ED, Stages of

virtual reality-based avatar intervention plus treatment as usual (TAU) or TAU. All participants will be assessed at baseline, at treatment cessation (12 weeks), and at 24 weeks post baseline. A stratified block-randomization with concealed randomization sequence will be conducted.

## Discussion:

A case-series study has demonstrated that a non-immersive (2D) application of avatar-based therapy is feasible and acceptable for patients with an eating disorder. While this preliminary evidence is promising, further research is needed to evaluate the efficacy of an avatar-based intervention for eating disorders. This current study will be the first investigating this by testing a 3D immersive version of the intervention in a large-scale, methodologically rigorous trial. Should the efficacy of this intervention be confirmed, it could open new avenues for research into psychological treatments for eating disorders.

## Trial registration

ClinicalTrials.gov NCT06345040

## Introduction

### Background and rationale

Eating disorders are characterized by dangerous eating habits and a tendency to overestimate the importance of weight and body shape in one's self-evaluation [1]. These illnesses affect at least 7% [2] of the global population and have profound physical and psychological costs for the affected individual [3], involving a high risk of early mortality [4,5]. Additionally, eating disorders are associated with a significant economic burden, including health care costs, reduced employment rates, and loss of productivity. In the European Union the annual costs approaches €1 trillion, exceeding the economic impact of anxiety and depression [6].

First line treatments for eating disorders include psychological therapies [7], however, many psychological interventions for eating disorders report suboptimal outcomes [8–10] and dropout rates have been estimated to range from 20–70% [11,12]. Even when treatment is completed, relapse rates have been found to be around 30% [12–14]. Consequently, there is significant interest in developing novel therapies that are engaging and capable of addressing characteristics that may serve as maintaining factors in eating disorder psychopathology.

**The inner voice of the eating disorder, a new target of intervention.** An estimated 94% of patients with an eating disorder report a critical internal voice commenting in second or third person on shape/weight, eating, and self-worth, often referred to as the 'eating disorder voice' [15,16]. Patients report that the eating disorder voice emerges at the onset of their illness. It is commonly perceived as supportive initially; however over time, it transforms into a critical and dominant voice, forcing the individual to use destructive eating behaviors [16].

Change in Eating Disorders Questionnaire; SSQ, Simulator Sickness Questionnaire; TAU, Treatment As Usual; VAAS, Voices Acceptance Scale; VPDS, The Voice Power Differential Scale; VR, Virtual Reality.

Qualitative studies have explained the eating disorder voice's transformation into a critical and dominating presence as a shift in the relationship between the self and the eating disorder. Initially valued by the person, the disorder becomes a more dissociated and stressful part of the self that overrules the desires and qualities of the 'true' self, leaving the individual feeling powerless [17]–[19].

The experience of a more powerful eating disorder voice is associated with increased use of compensatory behaviors (e.g., fasting, vomiting, laxative misuse, compensatory exercise) and longer illness duration [20]. In line with this, research suggests that the individual's subordinate position to the eating disorder voice could play a substantial role in maintaining eating disorder symptoms and contribute to challenges in treatment engagement [20]. Despite its clinical relevance, this power dynamic is rarely addressed directly in standard treatments. Targeting it in therapy may enhance clinical outcomes and treatment engagement by helping patients regain control over the voice and strengthen their identity beyond the disorder [16,18,21,22].

**Avatar-based therapy for eating disorders.** A growing body of research suggests that the experience of an inner eating disorder voice shares a key relational feature with distressing verbal auditory hallucinations, namely the subordinate position that individuals often adopt in relation to the voice [18,20]–[22]. This overlap has prompted interest in applying relational therapies developed for voice-hearing (e.g., voice dialogue), to address the eating disorder voice, an approach supported by emerging qualitative findings [18,22].

AVATAR Therapy, a novel relational intervention, originally designed for voice hearing targets this shared power imbalance. In the therapy, participants engage in real-time dialogue with a digital avatar representing their distressing voice, allowing them to challenge the voice's position. The intervention has demonstrated efficacy in reducing the severity of the psychotic voice and alleviating associated distress within two fully powered randomized controlled trials [23,24]. Given that AVATAR Therapy directly targets the relational dynamic shared between voice hearers and people with eating disorders, it provides a compelling rationale for testing the intervention's applicability in eating disorder populations [18,25].

Based on this, a case-series feasibility study involving 12 participants, tested the application of AVATAR therapy for patients with anorexia nervosa, with the aim of helping participants regain power over their eating disorder voice and reconnect with their identity beyond the eating disorder [17]. The treatment approach met the predetermined thresholds for both acceptability and safety [17]. Furthermore, while not powered as a test of efficacy, the study provided promising indications of potential efficacy in enhancing self-compassion and reducing symptoms of stress and anxiety [17]. Qualitative data from the study suggested that participants reported increased confidence in asserting themselves against the eating disorder, which contributed to a sense of empowerment to resist its influence [17].

The first AVATAR therapy for psychosis and anorexia nervosa was delivered on a 2D computer screen. Our research group recently tested the efficacy of an immersive 3D avatar intervention for psychosis, mirroring the positive findings from the original

AVATAR therapy trial [26]. Building on this, we adapted the virtual reality (VR) software to individuals with eating disorders. In a feasibility study (n = 10), this immersive version met the predefined thresholds for feasibility and acceptability and showed promising preliminary treatment effect (unpublished work). Based on these initial findings, the current study aims to investigate the efficacy of an immersive VR-based avatar intervention for eating disorders. The study represents the first fully powered, large-scale, methodologically rigorous clinical trial to explore whether a relational, dialogue-based approach, such as avatar intervention can reduce symptoms and improve quality of life for patients with an eating disorder.

The application of avatar-based intervention to eating disorders is an innovative approach that addresses a critical and underexplored mechanism in eating disorder pathology. By giving form and voice to the internal critical dialogue, avatar-based intervention may facilitate a clearer separation between the individual and the disorder, an important mechanism in the context of eating disorders which are often ego-syntonic and characterized by ambivalence towards change [21,27]. The immersive format of VR-based avatar intervention may support the development of a stronger, recovery-oriented self-narrative; something that traditional talking therapies may struggle to achieve with the same emotional impact. If the study yields positive findings, it could pave the way for new research on psychological interventions for eating disorders. The next step would involve replicating these results across diverse settings.

### Objectives

- **Primary objective:** To evaluate whether a VR-based avatar intervention combined with treatment as usual (TAU) results in greater reductions in eating disorder symptoms compared to TAU alone, as measured by the global score on the Eating Disorder Examination Questionnaire (EDE-Q). A clinically meaningful difference is defined as a between-group difference of ≥0.83 on the EDE-Q global score.

- **Secondary objectives**: To evaluate whether the VR-based avatar intervention plus treatment as usual (TAU), compared to TAU alone, leads to greater improvements in cognitive and behavioral domains associated with the eating disorder voice experience and eating disorder psychopathology. These domains include voice appraisal (e.g., perceived power, omnipotence, and malevolence of the eating disorder voice), identification with the eating disorder, level of depressive symptoms, and motivation for change (e.g., readiness and confidence to recover). Each domain will be assessed using validated self-report measures, and between-group differences will be evaluated at post-intervention.

- **Exploratory objectives:** To examine the effects of the VR-based avatar intervention plus TAU on a broader range of cognitive, affective, behavioral outcomes related to eating disorder psychopathology. These include measures of body image, self-compassion, emotion regulation and cognitive flexibility. Full details of all exploratory outcomes, including timing and measures, are provided in the section 'Exploratory Outcomes'.

### Trial design

The Dialogue study is a randomized parallel-group superiority clinical trial allocating a total of 96 patients to either VR-based avatar intervention plus treatment as usual (TAU) or TAU (see Table 1 for participant eligibility criteria) Participants will be randomly assigned to the intervention plus treatment as usual and TAU in a 1:1 ratio (48 participants in each group). As recruitment will take place from different treatment sites, TAU may consist of various forms of treatment courses, all provided by health professionals. These treatments may include individual or group psychotherapy, and for some participants, complementary dietary guidance and/or meal support, depending on the severity of eating disorder symptoms (for further description of TAU, see the section 'Explanation for the choice of comparators'). All participants will be assessed at baseline and at 12- and 24 weeks post baseline (see Fig 1). A stratified block-randomization with concealed randomization sequence will be conducted.

**Table 1. Eligibility criteria**

| Inclusion Criteria: |
|---|
| 1. ≥Age 18 of years |
| 2. Ability to give informed consent |
| 3. A diagnosis of eating disorder: Anorexia Nervosa, Bulimia Nervosa, Atypical Anorexia Nervosa, Atypical Bulimia Nervosa (ICD-10 code: F50.00 – F.50.30) |
| 4. A global score ≥ 2.77 on the Eating Disorder Examination Questionnaire (EDE-Q) |
| 5. Recognizing having an internal eating disorder voice |

| Exclusion Criteria: |
|---|
| 1. Organic brain disease |
| 2. Comorbid psychosis (ICD-10 code: F20.00-F29.00) |
| 3. Psychotic depression (ICD-10 code: F33.30-33.31) |
| 4. Immediate risk of suicide |
| 5. Risk of refeeding syndrome |
| 6. A command of spoken Danish or English inadequate for engaging in therapy |

## Ethics approval and consent to participate

The study has been approved by the Committee on Health Research Ethics of the Capital Region of Denmark (reference number: H-22067692) (see S1 File: Latest protocol approved by ethics committee) and the Danish Data Agency (reference number: P-2022–932). Written informed consent will be obtained from all participants in the study. Participants have the right to withdraw from the trial at any time due to any reason. Participants will be asked whether they wish to have their data deleted upon withdrawal. If they do not request deletion, data collected up to the point of withdrawal will be retained and used in analyses, in accordance with ethical guidelines and national data protection regulations. If data remain linkable (e.g., via a pseudonym or code), it may be deleted upon request prior to publication. However, once data are fully anonymized or shared in de-identified form with third parties, individual data can no longer be withdrawn.

## Methods: Participants, interventions, and outcomes

### Study setting

Recruitment will take place from eating-disorder outpatient facilities in the Mental Health Services in the Capital Region of Denmark and in Region of Zealand of Denmark, as well as private hospitals, residential treatment sites, private practice psychologists, and interest groups. The randomized clinical trial will be conducted at the University Hospital at Mental Health Center Copenhagen, Capital Region of Denmark. The SPIRIT 2013 reporting guidelines has been applied together with the SPIRIT 2025 explanation and elaboration: updated guideline for protocols of randomized trials [28,29]. A completed SPIRIT checklist has been provided as an additional file (see S2 File: SPIRIT checklist).

### Who will take informed consent?

Initially participants will receive information about the study by referring clinicians and in handouts accessible, for instance, in the outpatient- and private facilities. Then information on the study is outlined in a telephone-call with an assessor from the Dialogue research group, and finally participants receive written information by e-mail before baseline assessment (e.g., information about study design, potential side effects, and participant rights). The assessor will discuss the above listed information with the participant at the baseline interview and the participant will have the opportunity to have any questions addressed. If deciding to take part in the study, the participant will sign a consent form.

| | STUDY PERIOD | | | | | | | |
|---|---|---|---|---|---|---|---|---|
| | Enrolment | Allocation | Post-allocation | | | | | Close-out |
| TIMEPOINT | $-t_1$ | 0 | $t_1$ | $t_2$ | $t_3$ | $t_{4...}$ | $t_7$ | $t_{1\&2}$ |
| **ENROLMENT:** | | | | | | | | |
| **Eligibility screen** | ▓ | | | | | | | |
| **Informed consent** | ▓ | | | | | | | |
| **Allocation** | | ▓ | | | | | | |
| **INTERVENTIONS:** | | | | | | | | |
| *Treatment as usual (TAU)* | | | ▓ | ▓ | ▓ | ▓ | ▓ | |
| *VR-based avatar intervention + TAU* | | | ▓ | ▓ | ▓ | ▓ | ▓ | |
| **ASSESSMENTS:** | | | | | | | | |
| *Baseline variables:* | ▓ | | | | | | | |
| **Primary outcome:** | | | | | | | | |
| EDE-Q | | | | | | | | |
| **Secondary outcomes:** | | | | | | | | |
| EAVE-Q, BAVQR, HADS, | | | | | | | | |
| IDEA, SOCQ-ED | | | | | | | | |
| **Exploratory outcomes:** | | | | | | | | |
| BSQ-34, EDQLS, P-CED, | | | | | | | | |
| VPDS, VAAS, PSYRATS-AH, | | | | | | | | |
| SCS, CSQ, DAS-24, ERQ, | | | | | | | | |
| GSE, BRIEF, Dflex, QPR-15 | | | | | | | | |
| *Outcome variables:* | | | | | | | | ▓ |
| **Primary outcome:** | | | | | | | | |
| EDE-Q | | | | | | | | |
| **Secondary outcomes:** | | | | | | | | |
| EAVE-Q, BAVQR, HADS, | | | | | | | | |
| IDEA, SOCQ-ED | | | | | | | | |
| **Exploratory outcomes:** | | | | | | | | |
| BSQ-34, EDQLS, P-CED, | | | | | | | | |
| VPDS, VAAS, PSYRATS-AH, | | | | | | | | |
| SCS, CSQ, DAS-24, ERQ, | | | | | | | | |
| GSE, BRIEF, Dflex, QPR-15 | | | | | | | | |
| *Potential moderators of treatment outcome:* | ▓ | | | | | | | |
| MINI, CTQ, SSQ | | | | | | | | |
| MPS | | | x | x | x | x | x | |

**Fig 1. Schedule of enrolment, interventions, and assessments.**

## Additional consent provisions for collection and use of participant data and biological specimens

Participants will be provided with information regarding the collection and use of their data and biological specimens. Participants will be asked to give permission for data to be stored for possible future research related to the current study. Information on storage duration, security measures, and policies for data and specimen destruction will be provided.

Participants can withdraw their consent at any time. Conditions for sharing data and specimens with third parties, including anonymization procedures, will be explained. Options for receiving individual and aggregate research results will be outlined.

## Interventions

**Explanation for the choice of comparators.** Eating disorder outpatient treatment within the Mental Health Services in Denmark consists of highly specialized interventions delivered by interdisciplinary health professionals. Depending on the severity or chronicity of the symptoms, treatment may include individual personalized psychotherapy and dietary counselling without a predetermined time limit, or standardized group therapy (lasting 20 weeks), which includes some sessions with a dietitian. Treatment in private hospitals generally involves 10 sessions of individual psychotherapy and dietary counselling, with the possibility of extension, while residential treatment facilities specializing in eating disorders offer a combination of individual psychotherapy, dietary counselling, and meal support throughout the duration of the stay. In the primary sector, treatment for eating disorders predominantly consists of individual psychological intervention, which may be complemented by dietary guidance. Some patients also receive meal support in their own homes, either as an adjunct to their treatment in the mental health services or as the primary form of support, typically provided by the municipality or private clinics. As is typical for most short-term psychological interventions for eating disorders, VR-based avatar intervention would likely not be implemented as a standalone treatment in clinical practice, particularly within the mental health services. Instead, it would be integrated into a broader treatment regimen. Consequently, it is most relevant to investigate its efficacy as an adjunctive treatment to determine its performance in actual clinical settings. Treatment as usual (TAU) would serve as the appropriate comparator to evaluate the VR intervention's unique contribution to patient outcomes, as it represents the current standard of care across treatment sectors, including the mental health services. Recruitment has proceeded more slowly than anticipated under our original inclusion criteria. Consequently, we have broadened eligibility and expanded the TAU comparator to include support from private dietitians and consultations with private psychiatrists or general practitioners. This protocol amendment will clearly be documented in the reporting of the trial results.

**Intervention description.** In the experimental condition, participants will receive seven individual sessions of a VR-based avatar intervention, conducted by trained psychologists with experience in this method. The therapists involved in the trial have previously delivered VR-based avatar interventions in our research group's study on VR-based avatar intervention for psychosis [26]. Therapists will be supervised monthly by Dr. Thomas Ward, with expertise in avatar-based therapy and Prof. Valentina Cardi, specialized in eating disorder treatment, who collaborated on the feasibility study on AVATAR Therapy for eating disorders.

The therapy is conducted in a shared setting, with both the therapist and participant present in the room, utilizing a computer and VR headset to deliver the VR part of the intervention. During the initial session, participants and the therapist will conduct a comprehensive assessment of the participants' eating disorder voice, capturing key distressing verbatim content and distinctive characteristics. Subsequently, participants, in collaboration with the therapist, will create a computer-generated avatar embodying their eating disorder voice. Additionally, a voice modulation program will be used to adjust the therapist's voice to match the imagined tone and pitch of the "eating disorder voice", as perceived by the participant. In the following sessions, the therapist initiates a dialogue between the participant and the avatar by shifting between talking as the avatar or as a supportive therapist. The participants are encouraged to confront the avatar with the aim of gaining increased control over their eating disorder voice.

To mirror the dynamics inherent in daily life between participant and their eating disorder voice, the avatar maintains a consistent stance on pertinent topics (e.g., food, training, body image). This in turn gives the participants the opportunity to articulate their own perspectives, tackle ambivalence, and assert their personal beliefs and values. As the therapy course

evolves, the avatar gradually diminishes its power/concedes aligning with the participant's increasing resilience against the influence of the eating disorder voice.

Initially, the dialogue will rely on pre-arranged responses from the participants and verbatim statements from their eating disorder voice. With the progression of therapy, the dialogue evolves into a more spontaneous exchange to enhance immersion. Throughout this process, the therapist monitors distress and checks in with the participants, adjusting the avatar dialogue accordingly. Each therapy session lasts 60 minutes of which approximately 15 minutes is spent in dialogue with the avatar. The remaining time involves preparing the participants for the avatar confrontation and evaluating the interaction. Generally, the therapist offers non-judgmental acceptance and positive regard, fostering self-exploration and self-acceptance [30].

When engaging with the avatar, participants will wear a VR headset and noise cancelling headphones, immersing themselves in a virtual living room where the avatar sits in front of them behind a desk. The software allows the therapist to control the proximity of the avatar, as a means of grading the exposure. Virtual food can be introduced into the environment, either to trigger the eating disorder voice or to challenge it in situations where its impact is typically potent [31], enhancing the transferability of the intervention [32,33]. The software, developed by Khora-VR, is based on the program used in our trial, applying virtual avatars in the treatment of psychosis [26]. The software has been modified to match the target group (e.g., body figure of the avatar) with the involvement of people with lived experiences (patients with eating disorders). Experiences from our ongoing trial on virtual reality-based intervention for psychosis shows this VR-program to be well-tolerated [26].

As part of the intervention, the participants are encouraged to take a picture of the avatar and are given audio recordings of the avatar-dialogues. They are then asked to listen to the recordings between sessions to further bolster their sense of control/autonomy and strengthen motivation for change. In addition, these recordings can be used by participants post-intervention to help sustain progress and potentially reduce the risk of relapse. The recordings also enable participants to share their inner experience of having an eating disorder voice with family, friends, and treatment providers.

### Fidelity

All therapy sessions will be audio recorded and a total of ten therapy courses will be randomly selected for fidelity rating by independent raters. The fidelity rating manual is an adaptation of the manual from our trial on VR-based avatar intervention for psychosis, modified to address topics related to the eating disorder voice. Fidelity rating will encompass adherence to the treatment manual as well as quality rating of general therapeutic skills (e.g., empathy, appropriate responding, handling of person specific aspects (e.g., cultural background, sex, history of trauma).

### Criteria for discontinuing or modifying allocated interventions

Participants can choose to discontinue treatment at any time. It is made clear that they will be able to do so without any impact on their standard care. Treatment and post-baseline assessments may be postponed due to participant's deterioration of mental health or hospitalization. If deterioration may be a consequence of the therapy the case will be dealt with in supervision, and it will be evaluated whether therapy needs to be terminated. Therapists can also contact the clinicians overseeing TAU to discuss further action. Any psychiatric admission and number of hospitalization days will be mapped upon trial completion.

### Strategies to improve adherence to interventions

Improvement of adherence is considered integral to the ongoing dialogue between the participant and therapist. Additionally, trial therapists may consult with the clinicians overseeing TAU to discuss appropriate subsequent

actions. If participants in the experimental condition experience fear of entering the virtual environment or confronting the avatar, the therapist can employ various strategies to mitigate these concerns (e.g., exposure to the computer screen instead of the full VR experience or graded proximity to the avatar in VR (see "intervention description").

### Relevant concomitant care permitted or prohibited during the trial

N/A: No special provisions.

### Provisions for post-trial care

Participants will receive their standard care before, during and after trial intervention.

### Outcomes

**Eligibility screening and primary treatment outcome.** Eligibility will be determined by a global score greater than or equal to 2.77 on the Eating Disorder Examination Questionnaire (EDE-Q), to ensure participants included in the study show clinically significant eating disorder symptoms. Eligibility will further be determined by a baseline interview assessing participants' experience of an inner 'eating disorder voice'. For participants without a formal eating disorder diagnosis from the mental health services, a diagnostic interview will be conducted by a trial assessor during the baseline interview.

The primary outcome of the study is eating disorder symptoms measured with the EDE-Q at the end of treatment (at 12 weeks follow up). The EDE-Q is a validated and widely utilized self-report questionnaire assessing eating disorder psychopathology and behaviors. It comprises 22 items, rated according to a seven-point forced-choice format (0–6). The questionnaire provides information about eating disorders' central behavioral features (e.g., fasting, binge eating, vomiting, laxative misuse, excessive exercising), with higher scores reflecting greater symptom severity or frequency. The EDE-Q comprises the four subscales: restraint, eating, shape, and weight concern [34]. The primary outcome will be the global score on the EDE-Q.

**Secondary treatment outcomes:**

- The experience of the eating disorder voice at the end of treatment (at 12 weeks follow up) assessed with the Experience of An Anorexic Voice Questionnaire (EAVE-Q). The EAVE-Q is a self-report questionnaire containing 18 items covering the experience of the eating disorder voice. Each question is scored from 1 (strongly disagree) to 5 (strongly agree). Higher EAVE-Q score is associated with higher symptom load in terms of eating disorder symptoms [35].

- Level of engagement with the eating disorder voice at the end of treatment (at 12 weeks follow up) measured with the Revised Beliefs About Voices questionnaire (BAVQR), subscale Engagement. BAVQR is a self-report questionnaire with 5 subscales assessing the individual's beliefs about the power and the benevolence/malevolence of the voice and the emotional, and behavioral ways of responding to it. Participants indicate responses on a 4-point Likert scale ranging from disagree (0) to strongly agree (3). The Engagement subscale is divided into emotional and behavioral ways of reacting. Higher scores reflect more emotional and behavioral engagement [36].

- Level of depressive symptoms at the end of treatment (at 12 weeks follow up) measured with The Hospital Anxiety and Depression Scale (HADS), subscale Depression. HADS is a self-report questionnaire consisting of 14 items measuring the level of anxiety and depression. Items for anxiety and depression are scored separately on a scale from 0–3 with a score range of 0–21 for both subscales. The higher the score the higher symptom load [37].

- Level of identification with the eating disorder and level of embodiment at the end of treatment (at 12 weeks follow up) measured with the Identity and Eating Disorder Questionnaire (IDEA). IDEA is a 23 item self-report questionnaire

assessing identification with the eating order and the experience of being (dis)embodied. Items are rated on a 5-point Likert scale ranging from 0 (do not agree) to 4 (strongly agree). The total score is the average of all item ratings. The higher score the more identification with the eating disorder and the less experienced embodiment [38].

- Motivation for change as experienced by the participant at the end of treatment (at 12 weeks follow up) measured with the Stages of Change in Eating Disorders Questionnaire (SOCQ-ED). The SOCQ-ED comprises 13 items measuring motivation to change. For each item, the participant selects among seven different response options which reflects the possible stage of change (i.e., precontemplation, contemplation, preparation, action, maintenance, and termination). Items are rated on a 6-point Likert scale ranging from 1–7, one being the possibility to exclude an irrelevant symptom domain. The higher score the more motivation to change [39].

**Exploratory treatment outcomes.**

- Level of eating disorder symptoms at 24 weeks follow up, measured with the EDE-Q global score.

- The experience of the eating disorder voice at 24 weeks follow up, assessed with the EAVE-Q.

- Level of engagement with the eating disorder voice at 24 weeks follow up, measured with BAVQR, subscale Engagement.

- Level of depressive symptoms at 24weeks follow up, measured HADS, subscale Depression.

- Level of identification with the eating disorder and level of embodiment at post treatment at 24 weeks follow up, measured with IDEA.

- Motivation for change as experienced by the participant at post treatment at 24 weeks follow up, measured with SOCQ-ED.

- Total score of eating disorder voice characteristics at post treatment (12- and 24 weeks follow up), assessed with a modified version of the Psychotic Symptoms Rating Scale, Auditory Hallucinations (the subscale: PSYRATS-AH). PSYRATS-AH consists of interviewer administered scales on frequency and duration of auditory hallucinations as well as scales on degree of distress associated with them. The modified version assesses frequency and duration of the eating disorder voice as well as the associated distress. Each item is rated on 5-point scale (0–4). The total score of PSYRATS-AH ranges from 0–44 with higher scores reflecting higher frequency, duration, and distress [40].

  - Level of quality of life at post treatment (12- and 24 weeks follow up) measured with the Eating Disorder Quality of Life Scale (EDQLS). EDQLS measures 12 domains of eating disorder related quality of life (i.e., disease-specific quality of life). Consists of 40 items, each rated on a 5-point Likert scale from strongly disagree (1) to strongly agree (5). Higher scores indicate better/higher ED-related quality of life [41].

  - Perceived difference in the power between the eating disorder voice and the patient at post treatment (12- and 24 weeks follow up) measured with the Voice Power Differential Scale (VPDS). VPDS is a self-report questionnaire with seven items measuring auditory hallucinations in terms of the perceived difference in power between the psychotic voice and the voice hearer. We are using a revised version to determine the power-balance between the eating disorder patient and his/her eating disorder voice. Power components include power, strength, confidence, respect, ability to cause harm, superiority, and knowledge. Each component is rated on a 5-point scale (1–5). The higher score, the more power the eating disorder voice is perceived to have compared to the eating disorder patient [42].

  - Level of acceptance and action in relation to the eating disorder voice at post treatment (12- and 24 weeks follow up), measured with the Voices Acceptance Scale (VAAS). VAAS is a self-reporting 31 item questionnaire comprising three scales: Acceptance, Action, and a full scale. Responses are provided on a 5-point Likert scale ranging from

strongly disagree (0) to strongly agree (4) with higher scores reflecting higher levels of acceptance and action in relation to the eating disorder voice [43].

- Level of self-compassion at post treatment (12- and 24 weeks follow up) measured with the Self-Compassion Scale (SCS). SCS is a 26 item self-report measure examining how individuals typically behave towards themselves while struggling. It consists of 6 subscales (Self-Kindness, Self-Judgment, Common Humanity, Isolation, Mindfulness, Over-Identification). Participants indicate, on a 5-point Likert scale ranging from Almost Never (1) to Almost Always (5), how often they think and feel as described. Higher scores indicate greater self-compassion [44].

- Level of body dissatisfaction measured with the Body Shape Questionnaire (BSQ-34) at post treatment (12- and 24 weeks follow up). BSQ-34 is a 34 item self-report questionnaire measuring body dissatisfaction, including concerns about body shape. Body dissatisfaction is assessed across four dimensions: avoidance and social shame related to body exposure, dissatisfaction with the lower parts of the body, use of laxatives and vomiting to reduce body dissatisfaction, and cognitions and maladaptive behaviors to control weight. Questions are answered on a Likert scale ranging from never (1) to always (6). Higher scores indicate greater concern about body shape and greater body dissatisfaction [45].

- Level of perceived self-efficacy at post treatment (12- and 24 weeks follow up), measured with the General Self-Efficacy Scale (GSE). GSE consists of 10 items rated on a Likert scale ranging from 1 (not at all true) to 4 (exactly true). Higher scores indicate greater self-efficacy) [46].

- Ability to regulate emotions at post treatment (12- and 24 weeks follow up), measured with the Emotion Regulation Questionnaire (ERQ). ERQ is a 10 item self-report questionnaire measuring how the respondent controls (regulates and manages) their emotions in two different ways: Emotional experience (cognitive reappraisal) and emotional expression (expression suppression). Items are answered on a 7-point Likert scale ranging from 1 (completely disagree) to 7 (completely agree). The two subscales are scored separately. Higher scores indicate greater use of the specific emotion regulation strategy [47].

- Level of executive functioning at post treatment (12- and 24 weeks follow up), measured with Behavior Rating Inventory of Executive Functioning (BRIEF). BRIEF is a 75 item self-report measure assessing executive functions. Participants indicate whether a specific behavior is a problem on a 3-point Likert scale (0=Never, 1=Sometimes, 2=Often). The global score (General Executive Function) ranges from 0–225. Higher scores indicate more problems with executive functioning [48].

- Level of cognitive (in)flexibility post treatment (12- and 24 weeks follow up), measured with Detail and Flexibility Questionnaire in Eating Disorders (Dflex). Dflex is a 24 item self-report questionnaire developed to measure two cognitive thinking styles often reported in patients with an eating disorder: cognitive rigidity (inflexibility) and excessive attention to details. It contains two subscales: Cognitive Rigidity Subscale and Attention to Detail Subscale. Participants respond on a 6-point Likert scale ranging from Strongly Disagree (1) to Strongly Agree (6). The higher the score, the more rigidity/attention to detail) [49].

- Participant satisfaction with the treatment, measured with the Client Satisfaction Questionnaire (CSQ) at post treatment (12 weeks follow up). CSQ is an 8 item self-report measure on satisfaction. Responses are given on a 4-point Likert scale ranging from poor (1) to excellent (4). Higher scores indicate higher satisfaction [50].

- Monitoring and reporting of adverse and unwanted events from psychotherapy measured with the Negative Effects Questionnaire (NEQ) post treatment (at 12 weeks follow up). NEQ is a 32 item self-report questionnaire examining events and outcomes that negatively impact the participant (these can be related to circumstances in the individual's life as well as related to treatment). For each item, the participants indicate whether they experience the statement (yes/no). If yes, the participant rates how negatively the event/outcome affected them on a 5-point Likert scale from

0 (not at all) to 4 (very much). Then it is indicated whether the negative event/outcome is related to treatment or other circumstances. The higher score the more negative effects [51].

- Level of dysfunctional cognitive beliefs post treatment (12- and 24 weeks follow up), measured with the Dysfunctional Attitude Scale questionnaire (DAS-24). DAS-24 is a 24 item self-report measure on dysfunctional cognitive beliefs. Responds are given on a 7-point Likert scale ranging from totally disagree (1) to totally agree (7). Higher scores reflects higher levels of dysfunctional beliefs [52].

- Positive and negative aspects of eating disorders post treatment as perceived by the participant post treatment (12- and 24 weeks follow up), measured with the Pros and Cons of Eating Disorders questionnaire (P-CED). P-CED is a 62 item self-report questionnaire assessing the participants perception of the pros and cons of the eating disorder. Answers are given on a 5-point Likert scale ranging from strongly agree (1) to strongly disagree (5). The sum of the pros and cons respectively are used to form an Ambivalence Score indicating the weight between the two [53].

- Level of personal recovery at post treatment (12- and 24 weeks follow up), measured with the Questionnaire about Process of Recovery (QPR-15) [54]. QPR-15 is a 15 item self-report measure for determining the level of personal recovery. Answers are given on a 5-point Likert scale ranging from strongly disagree (0) to strongly agree (4). The higher score the higher level of perceived recovery [54].

**Potential moderators of therapy outcome.**

- Presence of comorbidity at baseline assessed with the Mini International Neuropsychiatric Interview (MINI). MINI is a structured diagnostic interview for the main psychiatric Axis-I diagnoses in Diagnostic and Statistical Manual of Mental Disorders, Fifth Edition (DSM-IV) and International Statistical Classification of Diseases and Related Health Problems, 10th Revision (ICD-10). Participants are asked questions on mental health problems that require a YES or NO answer. Interviewer based rating is done by marking either YES or NO for each symptom within in each diagnosis module. At the end of each module, there are diagnostic boxes indicating whether the diagnostic criteria are met or not [55].

- Level of immersion in VR measured with a modified version of the Multi Modal Presence Scale (MPS) after each VR-based therapy session. MPS modified version is a 10 item self-report scale measuring the level of immersion in VR, relevant to the dialogue intervention's virtual environment. Answers are given on a 5-point Likert scale ranging from totally disagree (1) to totally agree (5). The higher score the more immersion [56].

- Presence of childhood traumas at baseline measured with the Childhood Trauma Questionnaire (CTQ). CTQ is a 28 item self-reporting instrument for retrospective assessment of trauma. Answers are given on a 5-point Likert scale ranging from never (1) to very often (5). Higher scores indicates greater severity of childhood trauma [57].

- Level of simulator sickness measured after each VR-based therapy session, with the Simulator Sickness Questionnaire (SSQ). SSQ is a 16 item self-report measure for simulator sickness. Participants are asked to rate symptoms on a 4-point Likert scale (0–3). The higher score the more symptoms (administered in therapy) [58].

## Participant timeline

See S1 Fig: Participant timeline (SPIRIT figure).

## Sample size

Our primary hypothesis posits a difference in treatment effect between the two groups, as measured by the Eating Disorder Examination Questionnaire (EDE-Q). Previous research has established a reliable clinically significant change of 0.83 on the

EDE-Q [59], with a standard deviation (SD) of 1.31 [60]. We define the minimal clinically important difference as a true difference of 0.83 between the experimental and control group means. Calculating effect sizes reveals that to detect treatment difference using a t-test, with 80% power and a significance level of 0.05, a sample size of 40 subjects per group is required Attrition rates in RCT's investigating psychological treatment for eating disorders has been reported to range from 0 to 34.2% for studies with immediate follow-up and 2.4% to 21.4% for studies with three-month post intervention follow-up [61]. To account for potential attrition and deviations from the assumptions, we have increased the total sample size by 20% to 96 participants.

## Recruitment

Recruitment takes place in collaboration with eating disorder-outpatient facilities in the Mental Health Services, annually receiving a total of approximately 1200 patients, private facilities and clinics, and interest groups. Relevant staff will be informed of the study and the criteria for inclusion and encouraged to refer patients who shows interest in the project. To avoid recruitment bias, staff will receive clarifying information as needed (e.g., research team attends conferences at inclusion site, provides written information and invites clinicians to discuss potential referrals). Additionally, we intend to disseminate information regarding the study through social media platforms, and via the online platform trialtree.com, mitigating contact between research projects and potential participants, providing individuals with a diagnosis of eating disorder the opportunity to self-refer. All referrals will be screened for eligibility.

## Assignment of interventions: Allocation

**Sequence generation.** Patients giving written consent are randomly allocated to either VR-based avatar intervention or standard treatment. Randomization is performed by a centralized, concealed computer-generated randomization system in REDCap (see "Data management section"). Randomization will be stratified by recruitment site to ensure both groups have a comparable range of TAU interventions.

Block size is created by an independent statistician and will be unknown to the trial assessors, trial therapists and clinicians overseeing TAU. The randomized intervention allocation is concealed to assessors and to the statisticians performing the analysis until the analysis of the resulting data has been finalized.

**Concealment mechanism.** Research personnel who are not blinded to treatment allocation (e.g., therapists) have exclusive access to the randomization program in Redcap and will perform the randomization process of each participant. After completion of baseline assessment, trial assessors inform the responsible personnel about new participants by sending an e-mail via a secured e-mail system managed by the Mental Health Services in the Capital Region of Denmark.

**Implementation.** Allocation sequences are generated by study personnel with access to the randomization program in REDCap. Enrolment of each participant will be conducted by therapists in the study.

## Assignment of interventions: Blinding

**Who will be blinded.** Assessors performing outcome evaluations are blinded. Due to the type of intervention participants, therapists conducting the VR-based avatar intervention, and the clinicians managing TAU will not be blinded to treatment. Self-reported outcome measures, including the primary outcome, will not be subject to blinding as the participants responding the questionnaires are not blinded to treatment. Throughout the trial (e.g., before each follow up assessment) participants will be instructed not to reveal their treatment condition to assessors performing outcome evaluation. On site, blinding is assured by separating the location of the assessment from the location of the VR-therapy sessions. Analysts will be blind to treatment allocation in performing the statistical analyses and in drafting the conclusions on treatment X and Y.

**Procedure for unblinding if needed.** In case an assessor becomes unblinded, the assessment will be conducted by another assessor blind to the treatment. Emergency unblinding of trial assessors are not considered relevant as there is ongoing communication between participant, trial therapist and clinicians managing TAU.

## Data collection and management

**Plans for assessment and collection of outcomes.** Assessment will be conducted at three time points: at baseline, treatment cessation (12 weeks post baseline), and at 24 weeks post baseline. Psychologists will perform the assessments upon receiving training and continuous supervision by a psychologist specialized in eating disorder assessment.

**Plans to promote participant retention and complete follow-up.** Participants will be contacted by telephone and receive questionnaires digitally one week prior to scheduled follow up interviews. Participants receive a reminder of their appointment by text a month prior to their 12- and 24-weeks follow up. Assessors will be flexible and rearrange the scheduled visits if needed. Participants who are unable to attend follow up assessment due to an unstable condition can be assessed at home or, if hospitalized, at the ward. If a participant leave the intervention program prematurely, clinical- and functional assessments will be performed, if possible.

**Data management.** Participant data will be entered directly into the electronical CRF (Case Report Form) during the assessment interview, using the data entry system REDCap. REDCap is an electronic data capture tool hosted by CIMT in the Capital Region of Denmark [62]. If necessary, data collection will be done on paper and later entered digitally in REDCap. Data on paper is stored on trial site in a locked filing cabinet in a locked office. REDCap has a complete audit trail on all data transactions, detailed user rights and access control management complying with Danish data legislation (Databeskyttelsesforordningen). Data will be exported to software packages such as SPSS, SAS, Stata, R and put in logged folders on a network drive controlled by the Capital Region of Denmark, CIMT [62]. A data manager will oversee variables are being properly defined with variable- and value tags and algorithms will be kept in special files. Data will be examined carefully to identify errors in data entry.

**Confidentiality.** Patient referrals, as well as other patient related information, will be exchanged between clinicians managing TAU and research team via a secured e-mail system managed by the Mental Health Services in the Capital Region of Denmark. REDCap access is only granted to relevant research personnel and will be restricted through two factor authentication. Data on each participant is connected to a unique serial number and will be exported from REDCap without personal identifiers. Surveys from REDCap are sent digitally to participants through a secure public mailing system (e-boks), transferring person sensitive information between government agencies (including health care services) and citizens. When participants open the link to the REDCap surveys, the form is completed directly in REDCap, reducing the risk of data loss and leak.

**Plans for collection, laboratory evaluation and storage of biological specimens for genetic or molecular analysis in this trial/future use.** A sub study will explore potential biological changes resulting from the intervention. Blood samples from the trial participants will be collected to analyse plasma nutritional markers (e.g., vitamins, trace elements, metabolites, hormones, and proteins) as potential biomarkers of altered eating behaviours post intervention. The biological measures of the study are exploratory and will be reported separately.

**Blood collection and storage.** At baseline and 12 weeks follow-up (cessation of treatment) patients will be asked to give a blood sample. Blood (approximately 20 mL) is drawn from a cubital vein by standard needle by an authorized laboratory technician, nurse, or doctor. Blood is collected in standard blood tubes. Whole blood and plasma will be collected for further analysis. Biochemistry markers of nutrition including albumin, haemoglobin, mean corpuscular volume (MCV), ferritin, vitamin D, calcium, magnesium, phosphate, zinc, vitamin B12, folate, total lymphocyte count (TLC) will be analysed according to best clinical practice. Furthermore, hormonal, metabolite and protein panels will be measured as an exploratory outcome to identify known and unknown markers of nutritional health these will be done using immune-based and mass-spectrometry based approaches. The biological material will be kept in a research biobank at a secured freezer at Mental Health Center Copenhagen until analysed. The research biobank has been approved by 'Videnscenter for Datasikkerhed´ in line with current legislation.

## Statistical methods

**Statistical methods for primary and secondary outcomes.** Analyses will be performed on the ITT population for the primary and secondary endpoints, i.e., analyzing all randomized participants according to the arm they were randomized to. The analysis method will be ANCOVA of change from baseline with treatment as factor and baseline value as covariate at the corresponding post-baseline visit, i.e., month 12. Treatment difference point estimates will be reported together with 95% CIs and p-value. Missing values will be multiply imputed

For the exploratory endpoints, a modified ITT set will be used, requiring at least one post-baseline measurement for the inclusion in the analysis for each endpoint. The analysis method will be linear mixed effects model. Results will be reported for both post-baseline visits.

Details of model specifications, imputation methods & assumptions, sensitivity analyses and subgroup analyses will be provided in the statistical analysis plan (SAP), that will be made publicly available prior to data lock (i.e., before the dataset is finalized and analysis starts).

Results will be reported with a 95% confidence interval and p-value, using a 5% significance level.

Baseline demographic and clinical characteristics will be summarized for each group, using descriptive statistics (e.g., means, standard deviations, medians, and range for continuous variables, frequencies and proportions for categorical) to assess comparability between groups at baseline. Efficacy analysis will be conducted by a blinded and independent statistician with no contact to the participants in the trial. Reporting will follow CONSORT guidelines [63].

**Interim analyses.** N/A no interim analyses will be performed in this trial.

**Methods for additional analyses (e.g., subgroup analyses).** Details of subgroup analyses will be provided in the SAP, that will be made publicly available prior to data lock (i.e., before the dataset is finalized and analysis starts).

**Methods in analysis to handle protocol non-adherence and any statistical methods to handle missing data.** For the primary and secondary endpoints, missing data will be multiply imputed under missing at random (MAR). Sensitivity analyses will involve exploring departures from the MAR assumption, i.e., a tipping point analysis is planned to assess how far must the imputed values deviate from the MAR assumption before the conclusion of the primary analysis is overturned

For the exploratory endpoints, missing data will be implicitly handled by the linear mixed effects model.

**Plans to give access to the full protocol, participant level-data and statistical code.** We will deliver a completely deidentified data set to an appropriate data archive for sharing purposes no later than 5 years after finishing the last 24 weeks follow up interviews.

## Oversight and monitoring

**Composition of the coordinating center and trial steering committee.** The trial will be supervised by a project management group (PMG). The PMG will consist of the primary investigator of the study (PI), the research collaborators on one of the inclusion sites in the Mental Health Services in the Capital Region, the daily leader of the trial, and the principal therapist of the trial. The PMG will have monthly meetings during the recruitment period.

**Composition of the data monitoring committee, its role and reporting structure.** A Data Monitoring Committee (DMC) will not be set up. Adverse events will be monitored according to ethical standards and continuously reported to the Danish Health Research Ethics Committee, overseeing the trial. The Danish Health Research Ethics Committee have the authority to recommend pausing or terminating the trial.

**Adverse event reporting and harms.** The main known side-effect to VR-based interventions is cyber sickness, resembling common motion sickness. If encountered, it typically resolves after a few VR-sessions as tolerance develops. VR-based psychotherapy is generally well tolerated [64]. Hence, we do not expect any serious adverse events to occur. As this is an emotion focused intervention set in an immersive condition, we do expect that participants may experience distress

as a temporary part of the therapeutic process, especially in the first sessions. Participants will be informed of this enabling them to make informed decisions regarding participation and adherence, and to prepare them for managing any potential transitory reactions [65]. As detailed in the section 'Intervention description', the therapists will monitor the participants' levels of distress throughout the therapy course, adjusting the intervention accordingly. If a participant becomes overwhelmed, therapists will assist with in-session and at-home calming strategies and follow up the next day by phone or text. Participants will also continue receiving standard care before, during, and after the intervention to support safety.

Any acute events impacting emotional or physical health will initiate immediate action, including contacting the participant's primary clinician, notifying relatives, or escorting the participant to acute psychiatric care, as appropriate. The following serious adverse advents are monitored throughout the study: 1) hospital admissions; 2) suicide attempts; 3) violent incidents necessitating police involvement (whether victim or accused); 4) self-harm; 5) deaths.

**Frequency and plans for auditing trial conduct.**  The PI of this study is a member of the PMG thus making sure information concerning trial conduct reaches members in the research group. Information regarding the trial is shared monthly.

**Plans for communicating important protocol amendments to relevant parties (e.g., trial participants, ethical committees).**  All protocol amendments need to be approved by the Committee on Health Research Ethics of the Capital Region of Denmark. Deviations from the published protocol will be documented in the trial registration on ClinicalTrials. gov (Identifier: NCT06345040).

**Plans for, patient or public involvement in the design, conduct, and reporting of the trial.**  The VR-based avatar intervention of this study was initially co-developed with individuals with a diagnosis of eating disorder (n = 8) and clinicians with expertise in eating disorder treatment (n = 5), whose input shaped the VR content and therapeutic manual to ensure relevance and acceptability. Additionally, participants in the feasibility study (n = 10) delivered feedback on the intervention, further informing the treatment protocol. Feedback on the experience of completing the assessment battery was also provided by an individual with lived experience. The analysis and dissemination phases will also include participants in the trial, ensuring their perspectives in the interpretation and communication of the findings.

**Dissemination plans.**  All results positive, negative, or non-significant will be published. Study results will be disseminated to the scientific audience, the public and trial participants, who in their written consent, have declared a wish to be informed. Discussions regarding the study and its results will be shared in scientific publications, conference presentations, clinical interest groups but also importantly in relevant patient interest groups.

**The use of AI.**  Generative AI (OpenAI's ChatGPT) was used as a writing support tool during the preparation of the protocol. The tool was used primarily to support reference searching during protocol development and for language editing and restructuring of text to improve clarity. All substantive ideas, research design, hypotheses, and technical and clinical details were developed by the research team and collaborators. All references and scientific claims have been manually verified. No text has been included without review, editing, and approval by the research team. The use of AI has been limited to support drafting and formatting, not content generation or decision-making.

**Trial status.**

- Protocol version and date: 1, 02.04.24

- Recruitment start: 26.02.24

- Clinical Trials registration: 14.03.2024 (within 21 days of enrollment of the first participant, in compliance with section 801 of the Food and Drug Administration Amendments Act (FDAAA 801), to ensure transparency)

- Estimated recruitment end: 26.10.25

- Estimated completion of data collection: 26.04.26

- Estimated date of results: 01.11.2026

## Discussion

As stated, eating disorder interventions are recommended to include psychological therapies [7], but in many cases these yield insufficient outcomes [8–10],with dropout rates ranging from 20–70% [11,12] and relapse rates of 30% upon completion [12–14]. This highlights a need for the development of therapies aimed at targeting characteristics that may maintain eating disorder symptoms. As research suggests that a subordinate relationship with the eating disorder voice, a critical inner voice experienced by patients with an eating disorder, may play a maintaining role in eating disorder pathology, this may present a meaningful target for intervention [16,18,21–22]. AVATAR therapy for psychosis has proven effective in reducing the power of a psychotic voice, hereby alleviating the voice hearer's distress [23,25]. A pilot study has provided preliminary evidence on the feasibility and acceptability of applying this computerized intervention to address the power dynamic between patients with an eating disorder and their inner eating disorder voice [17]. This trial explores the efficacy of this intervention for eating disorders, using avatars in virtual reality.

**Embodied externalization and dialogue as an important mechanism of change.** Prominent interventions for eating disorders such as Cognitive-Behavioral Therapy, the Maudsley Model of Anorexia Nervosa Treatment for Adults, and Focal Psychodynamic Therapy, typically focus on challenging the 'irrationality' and 'unhelpfulness' of thought content [21]. However, these traditional approaches often overlook the patient's relationship with these thoughts [21]. In contrast, avatar-based therapy, incorporates therapeutic modalities such as relational and narrative therapies, which emphasize addressing the subordinate stance that patients often adopt towards these problematic thoughts or parts of the self. As evident within the frame-work of avatar-based therapy, relational approaches often utilize externalization techniques to enable patients to challenge the influence of these thoughts or parts of self by facilitating a dialogue with them [21,27].

The application of avatars or computer-generated representations expands upon techniques like the chair work method, a dialogue-based approach that externalizes a part of the self in a physical form and space [21,27]. The chair work technique involves using two chairs to represent different aspects of the self, with participants physically shifting between chairs to adopt the perspectives of each part in real time [22,18,27]. These more embodied approaches, has been shown to be particularly useful for patients, who like those with eating disorders, often over-identify with a specific part of the self, enhancing their ability to differentiate themselves from their inner experiences [21,27].

However, the chair work method still depends heavily on the participants' imagination to embody the different parts of the self. In contrast, employing avatars allows patients to engage in a dialogue with a part of self, as if it were an *external independent entity*. Findings in cognitive science, suggest that experiencing oneself acting on an external entity, rather than on an internal representation, enhances the feeling of agency (e.g., the ability to make independent choices and act on them) [66]. Building on this, avatars not only help patients differentiate themselves from their inner experiences but also promote a sense of autonomy, potentially offering a more effective means to create dissonance between the self and the eating disorder, and achieve greater sense of control.

Compared to traditional talking therapies avatar-based interventions also poses new possibilities and challenges in the therapeutic relationship. When the therapist voices the eating disorder's often abusive content, it requires considerable training in the method to ensure the dialogue remains therapeutically contained and it poses new ethical concerns (also see discussion on 'Ethical implications') [67]. On the other hand, patients with an eating disorder often show ambivalence towards recovery, rendering motivational work a central part of treatment. Avatar-based intervention may help mitigate a common difficulty in this treatment phase: the tendency for therapists to argue for change, unintentionally positioning patients to defend the eating disorder [68,69]. By externalizing the eating disorder through an avatar, the patient can respond to the disorder directly, from other parts of the self, replacing the oppositional dynamic between therapist and patient with internal dialogue.

**The potential of VR enhanced externalization.** Unlike traditional talking therapies, which relies on reflective insight, avatar-based interventions provide an experiential context where individuals engage directly with an embodied part of the self. This immersive format allows for real-time work on "hot" cognitive, emotional, and relational processes [67]. A newly published

study on AVATAR therapy for psychosis found that sense of presence in the avatar dialogue was associated with emotions such as control and that higher levels of presence were linked to greater reductions in auditory verbal hallucinations, suggesting that presence may be a key component of therapeutic outcomes in avatar-based therapy [70]. The current study represents a pioneering approach within externalization techniques, employing immersive technology to embody the eating disorder part of self. Virtual reality enables users to interact within a simulated environment as if they were physically present in that environment [66], making it possible for participants to experience a sense of being present with their eating disorder voice in the same physical space [71]. This heightened level of presence may offer advantages over traditional methods such as chair work or desktop-based avatar interactions, as increased presence has been associated with enhanced clinical outcomes.

**Ethical implications**    Avatar-based intervention for eating disorders may be emotionally demanding, as it involves engaging participants in dialogues with digital a representation of their often abusive eating disorder voice. While many participants in avatar-based therapy report empowerment and emotional relief, the approach carries ethical risks, particularly when therapists voice abusive or discriminatory content through the avatar [67]. As a result, patient safety has been prioritized both in the conduct of the trial and in the evaluation of the intervention. Trial therapists are required to assess each participant's psychological readiness, monitor for distress, ensure ongoing informed consent, and actively manage any negative effects, as detailed in the 'Adverse Events Reporting and Harms section'. In addition, the study will systematically assess participant reported negative outcomes, using the Negative Effects Questionnaire (NEQ).

**Contributions and limitations of the study.**  The current study will be the first to explore the efficacy of an avatar-based intervention for eating disorders, using avatars in an immersive 3D format. The intervention aligns with narrative and relational therapies, in which externalization of and dialogue with the problematic part of self is an important mechanism for change. It represents a novel and potentially more effective way of treating eating disorders by helping patients assert themselves against the every-day experience of the eating disorder voice in an immersive dialogue. The goal of the intervention is to assist patients in reconnecting with their identity beyond the eating disorder.

We seek to extend preliminary findings of avatar-based intervention for eating disorders by investigating the efficacy of this approach in a fully powered study. As the study is experimental, a broad line of explorative outcomes has been included, with the purpose of informing the direction of future studies. These outcomes include participants' experience of recovery post-intervention, level of emotion regulation, and level of cognitive flexibility. We have also included a range of measures on the power balance between the participants and their eating disorder voice, as well as eating disorder-related symptoms and potential negative effects of the therapy.

The strengths of the study must, however, be seen in the light of the potential limitations. The extensive line of exploratory outcomes may give cause to chance of multiplicity. Therefore, any exploratory findings will be interpreted with caution. In terms of the study design, it can be debated whether an active control group may help discern whether the observed effects truly are because of the VR-intervention's treatment mechanisms or simply the result of increased interaction with treatment providers. On the other hand, a non-active control group like a waiting list, can provide valuable insights into the natural progression of the condition without any intervention. Although VR-based interventions have demonstrated lasting effects in related fields, such as anxiety disorders [72], the current study does not include extended follow-up periods. As such, it cannot address the durability of therapeutic gains over time. Future research should incorporate long-term follow-up to evaluate the persistence of this VR-based treatment's effect. A heterogeneous target group across different clinical settings, can make the results more representative for a larger population, thereby enhancing the external validity. However, it introduces challenges related to data analysis, making it more complex and potentially obscuring the actual impact of the intervention by not accurately reflecting its performance in specific subgroups or in real-world clinical settings. As a means of ensuring a level of homogeneity, we have introduced a cut-off score on eating disorder symptoms (identified with the self-report measure EDE-Q). Optimally, performing subgroup analysis can elucidate how different demographic and clinical characteristics may influence treatment outcome.

In conclusion, the trial will provide important and comprehensive insight into the efficacy of a novel, innovative VR-based treatment for eating disorders.

## Supporting information

**S1 File. Latest protocol approved by ethics committee.** The latest version of the protocol approved by the Committee on Health Research Ethics of the Capital Region of Denmark (reference number: H-22067692).
(DOCX)

**S2 File. SPIRIT checklist.** Completed SPIRIT 2013 checklist indicating where each item is addressed in the protocol manuscript.
(DOCX)

**S1 Fig. Participant timeline (SPIRIT figure).** Schedule showing the timing of participant enrolment, interventions, and assessments. Includes all key study procedures and follow-up timepoints.
(DOC)

## Acknowledgments

The authors express their utmost gratitude to research assistant Linnea Rosalinde Køber and therapist Tina Højsgaard Tinglef for their valuable contributions to the study. Special thanks to research assistant Katrine Rasmussen for her significant contributions during the start-up phase. We are also grateful to Anna Buus Kristensen and Frederikke Ekas Wilken for their efforts in organizing recruitment at Psychiatric Center Ballerup, and to Anne Bryde Christensen for her role as the main organizer of recruitment at Psychiatric Center Ballerup. Additionally, we thank Nora Trompeter for her supervision on assessment [28].

## Author contributions

**Conceptualization:** Nina K. Hansen, Emma S. Ries, Thomas Ward, Valentina Cardi, Anne B. Christensen, Merete Nordentoft, Nadia Micali, Louise B. Glenthøj.

**Funding acquisition:** Louise B. Glenthøj.

**Methodology:** Nina K. Hansen, Thomas Ward, Valentina Cardi, Carsten Hjorthøj, Merete Nordentoft, Nadia Micali, Louise B. Glenthøj.

**Project administration:** Louise B. Glenthøj.

**Supervision:** Nadia Micali, Louise B. Glenthøj.

**Writing – original draft:** Nina K. Hansen.

**Writing – review & editing:** Thomas Ward, Valentina Cardi, Anne B. Christensen, Carsten Hjorthøj, Merete Nordentoft, Nadia Micali, Louise B. Glenthøj.

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
