## [Decision Letter · Decision Letter 0]

Dear Dr. Hansen,

Thank you for submitting your manuscript to PLOS ONE. After careful consideration, we feel that it has merit but does not fully meet PLOS ONE’s publication criteria as it currently stands. Therefore, we invite you to submit a revised version of the manuscript that addresses the points raised during the review process.

The reviewers have made comments, and please resubmit an updated version after major revisions.

We look forward to receiving your revised manuscript.

Kind regards,

Dr. Mohammad Mofatteh, PhD, MPH, MSc, PGCert, BSc (Hons), MB BCh (c)

Academic Editor

PLOS ONE

Journal Requirements:

5. Please ensure that you refer to Figure 2 in your text as, if accepted, production will need this reference to link the reader to the figure.

6. Please remove all personal information, ensure that the data shared are in accordance with participant consent, and re-upload a fully anonymized data set.

Additional Editor Comments :

The reviewers have made commenrs and I come to the concluion that a major revision would be suitable. Please make substantial changes to your manuscript and resubmit.

Reviewers' comments:

Reviewer's Responses to Questions

**Comments to the Author**

1. Does the manuscript provide a valid rationale for the proposed study, with clearly identified and justified research questions?

Reviewer #1: Yes

Reviewer #2: No

Reviewer #3: Yes

2. Is the protocol technically sound and planned in a manner that will lead to a meaningful outcome and allow testing the stated hypotheses?

Reviewer #1: Yes

Reviewer #2: No

Reviewer #3: Yes

3. Is the methodology feasible and described in sufficient detail to allow the work to be replicable?

Reviewer #1: Yes

Reviewer #2: No

Reviewer #3: Yes

4. Have the authors described where all data underlying the findings will be made available when the study is complete?

Reviewer #1: Yes

Reviewer #2: No

Reviewer #3: Yes

5. Is the manuscript presented in an intelligible fashion and written in standard English?

Reviewer #1: Yes

Reviewer #2: No

Reviewer #3: Yes

You may also provide optional suggestions and comments to authors that they might find helpful in planning their study.

Reviewer #1: This is a RCT interesting study assessing the effectitveness of a VR intervention to reducing eating disorder symptoms.

Some minor comments:

In Line 153 or in section 508 - can the authors note the allocation ratio between Intervention: TAU.

Whats the rationale for not also stratifying by eating disorder symptom severity at baseline incase by chance there is imbalance between groups and is this prognostic of the outcome? i.e eht EDE-Q?

Line 154- define ITT - as you have done in line 612 - then no need to repeat in 612 or remove ITT sentence in Line 154

When patients withdraw/discontinue for whatever reason - does the consent also allow for the researchers to use the data up to the point of withdrawal - this should be made explicit in the protocol?, as it says they can withdraw consent - but not clear when it comes to other aspects such data sharing with third parties.

In the sample size section- can you state the attrition %age as it looks to be aroung 15%.

In the statistical methods sections add reporting guidelines will follow CONSORT guidelines. Also need a few sentence aroung planed descriptive sumaasries of baseline characteristis, i.e by randomised group.

As this an open label study - was the SAP signed off prior to recruitment see ICH guidelines on this.

Have some brief mention of the planned primary analysis and that treatment effect will be reported along side 95%CI and associated p-value, i.e 5% as the significance level.

i.e mention of missing data- can you state which outcomes this relates to, since primary outcome is collected at a single time point and baseline (which I assume will be adjusted for in the models).

Reviewer #2: The study is using AVATAR therapy, originally designed for psychotic disorders, to treat eating disorders. However, it does not introduce a completely new method and needs to justify why this approach would be effective.

It assumes that the "eating disorder voice" is similar to voices in psychotic disorders, but this connection is not clearly proven. More evidence is needed to confirm that AVATAR therapy will work in the same way.

The research is based on a small study using a simple (2D) version of the therapy, which is not enough proof to justify testing a more advanced (3D) version in a large trial. More small-scale studies should be done first.

Virtual reality therapy requires costly equipment and training. The study does not explain how this will be managed in hospitals and clinics, making it difficult to implement widely.

The study does not discuss how long the benefits of the treatment will last. It should mention whether patients might return to old habits over time and how to prevent this.

There are not enough details about how the study will measure success. It should clearly define what changes in patients' thinking or behavior will be used to evaluate the therapy’s effectiveness.

Talking to an avatar of their eating disorder voice could be emotionally distressing for some patients. The study does not discuss possible negative effects or how they will be handled.

Reviewer #3: Dear Authors,

Thank you for the opportunity to review your protocol investigating the effect of a virtual reality-based intervention for eating disorders titled “The Dialogue Study: A randomized clinical trial evaluating the effectiveness of a virtual reality-based intervention plus treatment as usual versus treatment as usual for eating disorders”.

Your proposed protocol presents a novel and potentially impactful approach to addressing eating disorder pathology through VR-based avatar interventions. What follows are some suggested changes you may want to consider incorporating in outlining the reasoning for the trial and how you plan to address some of common challenges of RCTs in the analysis and reporting.

Introduction:

The economic burden of eating disorders could be supported with specific data or references (e.g., Schmidt et al., 2016). Similarly, further evidence on the role of the eating disorder voice in symptom maintenance and treatment resistance could be included (e.g., Aya et al., 2019)

Intervention Design and Mechanisms of Change:

The explanation of embodied externalization and dialogue as mechanisms of change is a key strength of your manuscript. The comparison to chair work techniques is particularly insightful, as it situates your approach within existing therapeutic frameworks while highlighting the advantages of VR immersion. You may wish to further discuss how factors such as presence, emotional engagement, and therapeutic alliance in the VR setting compares to more traditional interventions (e.g., Natali et al., 2023; Ward et al 2020). In addition to the discussing the potential long-term benefits of avatar-based interventions (e.g., Freeman et al., 2017).

Outcome Measures:

The selection of outcome measures is comprehensive and encompasses various dimensions of the disorder. However, aligning the outcome measures with internationally recognized standards (e.g., ICHOM: International Consortium for Health Outcomes Measurement) would enable further comparability and applicability of findings to the existing literature (see Austin et al., 2023).

Statistical Methods:

The use of linear mixed models with repeated measurements is an appropriate choice for analyzing longitudinal data, allowing for robust assessment of intervention effects over time. Your intention-to-treat approach ensures rigor in statistical analysis. However, given the commonly high dropout rates seen in eating disorder interventions, how do the authors anticipate dealing with missing data (e.g., multiple imputation) and any subsequent sensitivity analyses (e.g., to assess the robustness of imputed results). Given the heterogeneity of eating disorder populations across different clinical settings, the authors may want to consider outlining the potential for subgroup analyses.

References

Schmidt, U., Adan, R., Böhm, I., Campbell, I. C., Dingemans, A., Ehrlich, S., Elzakkers, I., Favaro, A., Giel, K., Harrison, A., Himmerich, H., Hoek, H. W., Herpertz-Dahlmann, B., Kas, M. J., Seitz, J., Smeets, P., Sternheim, L., Tenconi, E., van Elburg, A., van Furth, E., … Zipfel, S. (2016). Eating disorders: the big issue.The lancet. Psychiatry,3(4), 313–315. https://doi.org/10.1016/S2215-0366(16)00081-X

Aya, V., Ulusoy, K., & Cardi, V. (2019). A systematic review of the 'eating disorder voice' experience.International review of psychiatry (Abingdon, England),31(4), 347–366. https://doi.org/10.1080/09540261.2019.1593112

Natali, L., Ward, T., Rowlands, K., Aya, V., Treasure, J., & Cardi, V. (2023). Changes in the eating disorder voice over time and the association of voice characteristics at baseline with clinical symptoms in patients with anorexia nervosa. Clinical psychology & psychotherapy, 10.1002/cpp.2934. Advance online publication. https://doi.org/10.1002/cpp.2934

Ward, T., Rus-Calafell, M., Ramadhan, Z., Soumelidou, O., Fornells-Ambrojo, M., Garety, P., & Craig, T. K. J. (2020). AVATAR Therapy for Distressing Voices: A Comprehensive Account of Therapeutic Targets. Schizophrenia bulletin,46(5), 1038–1044. https://doi.org/10.1093/schbul/sbaa061

Austin, A., De Silva, U., Ilesanmi, C., Likitabhorn, T., Miller, I., Sousa Fialho, M. D. L., Austin, S. B., Caldwell, B., Chew, C. S. E., Chua, S. N., Dooley-Hash, S., Downs, J., El Khazen Hadati, C., Herpertz-Dahlmann, B., Lampert, J., Latzer, Y., Machado, P. P. P., Maguire, S., Malik, Moser, C.M., Myers, E., Pastor, I.R., Russell, J., Smolar, L., Steiger, H., Tan, E., Trujillo-Chi Vacuán, E., Tseng, M.M., van Furth, E.F., Wildes, J.E., Peat, C., Richmond, T.K.(2023). International consensus on patient-centred outcomes in eating disorders. The lancet. Psychiatry,10(12), 966–973. https://doi.org/10.1016/S2215-0366(23)00265-1

Freeman, D., Reeve, S., Robinson, A., Ehlers, A., Clark, D., Spanlang, B., & Slater, M. (2017). Virtual reality in the assessment, understanding, and treatment of mental health disorders. Psychological medicine, 47(14), 2393–2400. https://doi.org/10.1017/S003329171700040X

**Do you want your identity to be public for this peer review?** For information about this choice, including consent withdrawal, please see our Privacy Policy

Reviewer #1: No

Reviewer #2: No

Reviewer #3: No

---

## [Author Response · Author response to Decision Letter 1]

16 May 2025

Dear editor,

Thank you for reviewing our manuscript and for the invitation to submit a revised version. We have carefully addressed all the points raised during the editorial process and by the reviewers. We are grateful for their time and thoughtful feedback, which has helped us significantly strengthen the manuscript.

Kind regards,

Louise Birkedal Glenthøj, corresponding author.

We have corrected the following editorial and formatting comments, to align with PLOS ONE’s requirements.

Comments from the Editor:

1. Please ensure that your manuscript meets PLOS ONE’s style requirements, including those for file naming.

2. We note that the grant information you provided in the ‘Funding information’ and ‘Financial disclosure’ sections do not match. When you resubmit, please ensure that you provide the correct grant numbers for the rewards you received for your study in the ‘Funding information’ section.

5. Please ensure that you refer to Figure 2 in your text as, if accepted, production will need this reference to link the reader to the figure.

6. Please remove all personal information, ensure that the data shared are in accordance with participant consent, and re-upload a fully anonymized data set.

Below we outline our responses to the reviewer’s comments:

Reviewer one:

This is a RCT interesting study assessing the effectiveness of a VR intervention to reducing eating disorder symptoms.

Response:

Thank you for thoughtful evaluation of our study and for finding it interesting.

Some minor comments:

1. In Line 153 or in section 508 - can the authors note the allocation ratio between Intervention: TAU.

Response:

Thank you for this comment. We agree, the allocation ratio should be stated clearly.

The following paragraph has been added to the ‘Trial design’ section:

“Participants will be randomly assigned to the intervention plus TAU or TAU in a 1:1 ratio (48 participants in each group).”

2. What’s the rationale for not also stratifying by eating disorder symptom severity at baseline incase by chance there is imbalance between groups and is this prognostic of the outcome? i.e eht EDE-Q?

Response:

Thank you for raising this important point. We did consider stratifying by baseline eating disorder symptoms, however, we estimated that this would add another layer of complexity to the already difficult task of recruiting. Additionally, our randomization scheme already stratifies by site. Adding an additional stratification variable would increase the likelihood of half-empty randomization blocks, in effect resulting in imbalances within strata. These two factors led us to drop the stratification by severity.

3. Line 154- define ITT - as you have done in line 612 - then no need to repeat in 612 or remove ITT sentence in Line 154.

Response:

Thank you for this observation. The ITT sentence has been deleted in line 154.

4. When patients withdraw/discontinue for whatever reason - does the consent also allow for the researchers to use the data up to the point of withdrawal - this should be made explicit in the protocol?, as it says they can withdraw consent - but not clear when it comes to other aspects such data sharing with third parties.

Response:

Thank you for this helpful comment. We agree that this should be clearly stated in the protocol. Participants are informed during the consent process that they may withdraw from the study at any time without consequence (e.g., drop-out). The participants will be asked if they wish to have their data deleted upon withdrawal. If they do not request for their data to be deleted, data collected up to the point of withdrawal will be retained and used in analyses, in line with ethical guidelines and national data protection regulations. If data remain linkable (e.g., via a pseudonym or code), it may be deleted upon request prior to publication. However, once data are fully anonymized or shared in de-identified form with third parties, individual data can no longer be withdrawn.

We have clarified this in the ethics section that now explicitly states:

‘Participants will be asked whether they wish to have their data deleted upon withdrawal. If they do not request deletion, data collected up to the point of withdrawal will be retained and used in analyses, in accordance with ethical guidelines and national data protection regulations. If data remain linkable (e.g., via a pseudonym or code), it may be deleted upon request prior to publication. However, once data are fully anonymized or shared in de-identified form with third parties, individual data can no longer be withdrawn.’

5. In the sample size section- can you state the attrition %age as it looks to be aroung 15%.

Response:

We agree that it would be helpful to add the attrition %age. An argument for this percentage has also been added:

The following has been added to the sample size section:

‘Attrition rates in RCT’s investigating psychological treatment for eating disorders has been reported to range from 0 to 34.2% for studies with immediate follow-up and 2.4% to 21.4% for studies with three-month post intervention follow-up 61. To account for potential attrition and deviations from the assumptions, we have increased the total sample size by 20% to 96 participants.’

6. In the statistical methods sections add reporting guidelines will follow CONSORT guidelines. Also need a few sentence aroung planed descriptive sumaasries of baseline characteristis, i.e by randomised group.

Response:

Thank you for these helpful suggestions. It has been added to the manuscript that reporting will follow CONSORT guidelines, see ‘Statistical analysis’ section.

We have also added a description of the planned descriptive summaries of baseline characteristics. Specifically, we will present baseline demographic and clinical characteristics using descriptive statistics (e.g., means, standard deviations, medians, and proportions) for each group. This will allow us to assess comparability between groups at baseline.

These details have been added to the Statistical Analysis section:

‘‘Baseline demographic and clinical characteristics will be summarized for each group, using descriptive statistics (e.g., means, standard deviations, medians, and range for continuous variables, frequencies and proportions for categorical) to assess comparability between groups at baseline.’

7. As this an open label study - was the SAP signed off prior to recruitment see ICH guidelines on this.

Response:

Thank you for this important question. The statistical methods, as they are outlined in this manuscript, was signed off before recruitment and was uploaded to clinical trials within 21 days of enrollment of the first participant, in compliance with section 801 of the Food and Drug Administration Amendments Act (FDAAA 801). A more detailed SAP is currently in preparation and will be made publicly available before database lock (i.e., before the dataset is finalized and analysis starts).

8. Have some brief mention of the planned primary analysis and that treatment effect will be reported alongside 95%CI and associated p-value, i.e 5% as the significance level. i.e mention of missing data- can you state which outcomes this relates to, since primary outcome is collected at a single time point and baseline (which I assume will be adjusted for in the models).

Response:

Thank you for this helpful comment. We agree that it should be stated that effect will be reported alongside 95%CI and associated p-value, i.e. 5% as the significance level.

The planned primary analysis in the statistical methods section now reads as follows:

‘Analyses will be performed on the ITT population for the primary and secondary endpoints, i.e., analyzing all randomized participants according to the arm they were randomized to. The analysis method will be ANCOVA of change from baseline with treatment as factor and baseline value as covariate at the corresponding post-baseline visit, i.e., month 12. Treatment difference point estimates will be reported together with 95% CIs and p-value. Missing values will be multiply imputed. For the exploratory endpoints, a modified ITT set will be used, requiring at least one post-baseline measurement for the inclusion in the analysis for each endpoint. The analysis method will be linear mixed effects model. Results will be reported for both post-baseline visits. Details of model specifications, imputation methods & assumptions, sensitivity analyses and subgroup analyses will be provided in the statistical analysis plan (SAP), that will be made publicly available prior to data lock (i.e., before the dataset is finalized and analysis starts).’

The variable for the primary endpoint is collected at two post-baseline visits, i.e., month 12 and 24, but the primary endpoint is set at month 12. Any missingness in this outcome at month 12 for the primary analysis will be addressed using multiple imputation. That is, for the parameters of primary and secondary endpoints, which are evaluated at month 12, multiple imputation will be used. For the exploratory endpoints, missing data will be indirectly handled through mixed effects models.

These additions have been made in the Statistical Analysis section:

‘For the primary and secondary endpoints, missing data will be multiply imputed under MAR. Sensitivity analyses will involve exploring departures from the MAR assumption, i.e., a tipping point analysis is planned to assess how far must the imputed values deviate from the MAR assumption before the conclusion of the primary analysis is overturned.

For the exploratory endpoints, missing data will be implicitly handled by the linear mixed effects model.’

Reviewer 2:

The study is using AVATAR therapy, originally designed for psychotic disorders, to treat eating disorders. However, it does not introduce a completely new method and needs to justify why this approach would be effective.

Response:

Thank you for this valuable comment. While AVATAR therapy itself is an established intervention, its application to individuals with eating disorders, in a fully immersive format, represents a novel and clinically meaningful adaptation. Many individuals with an eating disorder report experiencing a persistent, critical, and controlling inner voice, that may reinforce disordered behaviors and impedes recovery. Despite its clinical relevance, this inner voice is rarely targeted directly in standard treatments. AVATAR therapy offers a structured and experiential way to externalize and interact with this voice offering a new pathway for therapeutic intervention. Externalization is particularly important in the context of eating disorders, which are often ego-syntonic and characterized by ambivalence towards change. By giving form and voice to the internal critical dialogue, AVATAR therapy may facilitate a clearer separation between the individual and the disorder. This may support the development of a stronger, recovery-oriented self-narrative; something that traditional talk therapies may struggle to achieve with the same immediacy or emotional impact. Thus, while the therapeutic modality is not entirely new, its application in this context is innovative and addresses a critical and underexplored mechanism in eating disorder pathology. We believe this adaptation holds promise for enhancing treatment outcomes and expanding the clinical utility of AVATAR therapy.

In the ‘background’ section of the manuscript, we have stressed the innovativeness of applying a fully immersive avatar intervention to eating disorders:

‘The experience of a more powerful eating disorder voice is associated with increased use of compensatory behaviors (e.g., fasting, vomiting, laxative misuse, compensatory exercise) and longer illness duration 20. In line with this, research suggests that the individual’s subordinate position to the eating disorder voice could play a substantial role in maintaining eating disorder symptoms and contribute to challenges in treatment engagement 20 Despite its clinical relevance, this power dynamic is rarely addressed directly in standard treatments. Targeting it in therapy may enhance clinical outcomes and treatment engagement by helping patients regain control over the voice and strengthen their identity beyond the disorder.’

Plus:

‘The first AVATAR Therapy for psychosis and anorexia nervosa was delivered on a 2D computer screen. Our research group recently tested the efficacy an immersive 3D avatar intervention for psychosis, mirroring the positive findings from the original AVATAR therapy trial27. Building on this, we adapted the virtual reality (VR) software to individuals with eating disorders. In a feasibility study (n=10), this immersive version met the predefined thresholds for feasibility and acceptability and showed promising preliminary treatment effect (unpublished work). Based on these initial findings, the current study aims to investigate the efficacy of an immersive VR-based avatar intervention for eating disorders. The study represents the first fully powered, large-scale, methodologically rigorous clinical trial to explore whether a relational, dialogue-based approach, such as avatar intervention can reduce symptoms and improve quality of life for patients with an eating disorder. The application of avatar-based intervention to eating disorders is an innovative approach that addresses a critical and underexplored mechanism in eating disorder pathology. By giving form and voice to the internal critical dialogue, avatar-based intervention may facilitate a clearer separation between the individual and the disorder, an important mechanism in the context of eating disorders which are often ego-syntonic and characterized by ambivalence towards change 21,28. The immersive format of VR-based avatar intervention may support the development of a stronger, recovery-oriented self-narrative; something that traditional talking therapies may struggle to achieve with the same emotional impact. If the study yields positive findings, it could pave the way for new research on psychological interventions for eating disorders. The next step would involve replicating these results across diverse settings.’

1. It assumes that the "eating disorder voice" is similar to voices in psychotic disorders, but this connection is not clearly proven. More evidence is needed to confirm that AVATAR therapy will work in the same way.

Response:

Thank you for highlighting this important point regarding the scientific rationale for the intervention. While there is no evidence that the “eating disorder voice” is identical to the voice experience in psychotic disorders in all respects, research does sugg

---

## [Editor Report · Decision Letter 1]

The Dialogue Study: Protocol for a randomized clinical trial evaluating the efficacy of a virtual reality-based psychotherapy plus treatment as usual versus treatment as usual for eating disorders

PONE-D-25-05703R1

Dear Dr. Hansen,

We’re pleased to inform you that your manuscript has been judged scientifically suitable for publication and will be formally accepted for publication once it meets all outstanding technical requirements.

Kind regards,

Mohammad Mofatteh, PhD, MPH, MSc, PGCert, BSc (Hons), MB BCh (c)

Academic Editor

PLOS ONE

Additional Editor Comments (optional):

The authors have responsed well to the reviewers' comments and the manuscript has gone under major update.
---

## [Editor Report · Acceptance letter]

PONE-D-25-05703R1

PLOS ONE

Dear Dr. Hansen,

I'm pleased to inform you that your manuscript has been deemed suitable for publication in PLOS ONE. Congratulations! Your manuscript is now being handed over to our production team.

Kind regards,

on behalf of

Dr. Mohammad Mofatteh

Academic Editor

PLOS ONE